# Effect of therapeutic hypothermia on renal and myocardial function in asphyxiated (near) term neonates: A systematic review and meta-analysis

**Maureen van Wincoop**[1], **Karen de Bijl-Marcus**[1]*, **Marc Lilien**[2], **Agnes van den Hoogen**[1], **Floris Groenendaal**[1]

1 Department of Neonatology, Wilhelmina Children's Hospital, University Medical Centre Utrecht and Utrecht University, Utrecht, The Netherlands, 2 Department of Pediatric Nephrology, Wilhelmina Children's Hospital, University Medical Centre Utrecht and Utrecht University, Utrecht, The Netherlands

☯ These authors contributed equally to this work.
* k.a.marcus@umcutrecht.nl

**Data Availability Statement:** All relevant data are within the manuscript and its Supporting Information files.

## Abstract

### Background

Therapeutic hypothermia (TH) is a well-established neuroprotective therapy applied in (near) term asphyxiated infants. However, little is known regarding the effects of TH on renal and/or myocardial function.

### Objectives

To describe the short- and long-term effects of TH on renal and myocardial function in asphyxiated (near) term neonates.

### Methods

An electronic search strategy incorporating MeSH terms and keywords was performed in October 2019 and updated in June 2020 using PubMed and Cochrane databases. Inclusion criteria consisted of a RCT or observational cohort design, intervention with TH in a setting of perinatal asphyxia and available long-term results on renal and myocardial function. We performed a meta-analysis and heterogeneity and sensitivity analyses using a random effects model. Subgroup analysis was performed on the method of cooling.

### Results

Of the 107 studies identified on renal function, 9 were included. None of the studies investigated the effects of TH on long-term renal function after perinatal asphyxia. The nine included studies described the effect of TH on the incidence of acute kidney injury (AKI) after perinatal asphyxia. Meta-analysis showed a significant difference between the incidence of AKI in neonates treated with TH compared to the control group (RR = 0.81; 95% CI 0.67–0.98; p = 0.03). No studies were found investigating the long-term effects of TH on

**Funding:** The authors received no specific funding for this work.

**Competing interests:** The authors have declared that no competing interests exist.

myocardial function after neonatal asphyxia. Possible short-term beneficial effects were presented in 4 out of 5 identified studies, as observed by significant reductions in cardiac biomarkers and less findings of myocardial dysfunction on ECG and cardiac ultrasound.

## Conclusions

TH in asphyxiated neonates reduces the incidence of AKI, an important risk factor for chronic kidney damage, and thus is potentially renoprotective. No studies were found on the long-term effects of TH on myocardial function. Short-term outcome studies suggest a cardioprotective effect.

## Introduction

Perinatal asphyxia poses harmful consequences for all fetal organs including brain, heart and kidney. The incidence of hypoxic-ischemic encephalopathy (HIE) in developed countries is 1 to 2 per 1000 term live births. HIE accounts for 23% of neonatal deaths worldwide. Moreover, 30% of the neonates with moderate HIE and 90% of the neonates with severe HIE develop severe long-term disabilities, including seizures, mental retardation and cerebral palsy [1–4].

Previous studies have demonstrated that whole-body therapeutic hypothermia (TH) is associated with long-term neuroprotection in full term neonates. Subsequently, this therapy is now the standard of care in developed countries [4].

The neuroprotective effect of TH is achieved as a result of a decrease in cerebral metabolism, reducing the accumulation of excitotoxic neurotransmitters, slowing down cell depolarization and suppressing oxygen free radical release and lipid peroxidation of cell membranes. Furthermore, this treatment also has a role in the suppression of apoptotic processes in the brain by inhibition of caspase enzymes. TH also reduces the release of pro-inflammatory interleukins and cytokines, resulting in suppression of microglial activation and thereby reducing direct neurotoxicity [4]. As it is suggested that the main pathogenic processes of brain damage and other organ damage are partly similar after asphyxia, there appears to be an equally sound rationale for the use of hypothermic treatment to protect other organs than the brain [5].

Perinatal asphyxia is associated with a decreased organ perfusion, and may result in multi-organ failure [2]. Multi-organ damage in the surviving neonates poses a high risk of severe chronic morbidities, such as chronic kidney disease (CKD), resulting in 42 million disability adjusted life years (DALY's) from perinatal asphyxia [6]. In this review, we will focus on the kidney and the heart, as both of these organ systems are known to be affected by perinatal asphyxia with potential long-term sequelae. The objective of this review is to determine the possible short-term and long-term beneficial effects of TH on renal and myocardial function in asphyxiated (near) term neonates.

## Methods

We report this systematic review in accordance with the Preferred Reporting of Items for Systematic Reviews and Meta-Analyses (PRISMA) checklist (S1 Table). No review protocol exists for this review.

### Search strategy

We conducted a literature search on PubMed and The Cochrane Library for studies examining the effects of TH in (near) term asphyxiated neonates on the heart and kidney. We also

performed manual searches of reference lists of studies and reviews. The search was performed in October 2019 and last updated on 28 June 2020. Search term keywords included: hypothermia, cooling, neonate, infant, newborn, asphyxia, hypoxic ischemic encephalopathy, hypoxia, ischemia, chronic kidney failure, chronic kidney disease, renal insufficiency, heart failure, myocardial dysfunction, long-term, prognosis, follow-up, development, outcome. Both Medical Subject Heading (MeSH) terms and text words were used. Two different searches on the effect of TH in asphyxiated neonates on both renal and myocardial function were performed, one on long-term effects and one on short-term effects to possibly identify studies on outcomes that indicated predictors of long-term outcome in case the "long-term" search term did not identify sufficient relevant studies. For short-term effects, we searched for studies performed in the first period of life. For long-term effects, we searched for studies performed after the first year of life. The search on long-term effects included keywords regarding 'long-term' and these were left out in the search on short-term effects. If sufficient relevant articles on perinatal asphyxia could not be found, we expanded our search to include other indications for TH, including near-drowning and aortic arch surgery. No restrictions were used on language and date of publication. The exact search strategy is included in S1 Text. The literature search was performed by one reviewer (M.W.) and when any doubt occurred assessed by a second researcher.

## Study selection and data extraction

Included studies met the following criteria: (1) RCT or observational cohort design, (2) intervention with TH, (3) evaluation of patients in a setting of perinatal asphyxia (or in a setting of near-drowning or aortic arch surgery if not sufficient articles on perinatal asphyxia can be found), (4) studies that provided outcome data on heart and kidney function, (5) long-term outcome or a valid marker of long-term outcome must be reported.

Information was extracted from each included study on: (1) study characteristics (including authors, study period, study design, location of study, number of participants), (2) patient characteristics (including severity of HIE), (3) the study's inclusion and exclusion criteria, (4) type of intervention (including cooling type, target temperature and period) and (5) outcome (including definition used/parameters assessed and time of measurement). The study selection and data extraction were performed by one researcher (M.W.) and when any doubt occurred assessed by a second researcher.

## Quality assessment

Two researchers (M.W., A.H.) assessed methodological quality of the identified RCT's using Cochrane's risk of bias [7]. Because it was impossible to achieve blinding in these studies, we did not include this criterion in the quality assessment. Each item of the risk of bias scored minus 1 point for high risk of bias, 0 points for unclear risk of bias and 1 point for low risk of bias, so that each article would get a total score ranging between -6 and 6 points. A score of 3 or lower was labelled high risk of bias and therefore 'low' quality, a score of 4 was labelled 'moderate' risk of bias and moderate quality and a score of 5 or 6 was labelled as low risk of bias and thus of 'high' quality. The National Institute of Health Quality Assessment Tool (NIHQAT), consisting of 14 items, was used to assess methodological quality for observational cohort studies (S2 Text). The tenth item was not applicable in these studies, therefore only 13 of the 14 items were included. A score of 11–13 was labelled 'high' quality, a score of 7–10 was labelled 'moderate' quality and a score ≤7 as 'low' quality [8]. 'Low' quality studies were not included in the meta-analysis. Rigour and trustworthiness was secured by assessing the

included studies independently by two researchers (M.W., A.H.). Any discrepancies in risk assessment were resolved by discussion until agreement was reached.

## Statistical analysis

If possible, a meta-analysis on the studies was performed. The statistical analysis was performed using Review Manager software (RevMan 5.3) [9], supplied by The Cochrane Collaboration. We calculated the risk ratio (RR) and risk difference (RD) for dichotomous data and the mean difference (MD) for continuous data, with 95% confidence intervals (CI) for all analyses. Subgroup analysis on the method of cooling was performed if relevant. As heterogeneity between the studies is likely due to small differences in e.g. study population, sample size, study quality and method, we used a DerSimonian and Laird random effects model. Heterogeneity of effects was measured with the statistic $I^2$ and the confidence intervals for $I^2$ were calculated [10]. A p-value of $<0.10$ was considered to be statistically significant heterogeneity. To investigate publication bias, Egger's test will be performed if $>10$ studies are included [7]. Sensitivity analysis will be performed using the leave-one-out method to assess how each individual study affects the overall estimate of the rest of the studies.

## Results

### Renal dysfunction

**Systematic search.** Database searching on long-term effects identified 69 citations. Five citations were duplicates and 64 studies were included for analysis. However, no relevant studies regarding the long-term effects of TH on the incidence and risk of developing CKD, defined as structural or functional abnormalities of the kidneys or a GFR $<60mL/min/1.73m^2$ for $\geq3$ months, after neonatal asphyxia were found [11]. Follow-up articles of the CoolCap, NICHD and TOBY trials did not include renal parameters [12–14]. Including keywords on other indications for TH in our search identified 13 more studies, but none of these were relevant.

Database searching on short-term effects identified 110 citations on the effect of TH on the incidence of acute kidney injury (AKI), previously called acute renal failure (ARF), after perinatal asphyxia and 6 more were identified by other sources, including reference lists [15]. A total of 107 citations were excluded because they were duplicates or the titles and abstracts or full texts were not relevant for this review. Overall, there were 9 studies included in this systematic review. A flow diagram is presented in Fig 1.

**Patient characteristics.** The nine included studies were carried out between 1996 and 2015. All 9 trials were randomized controlled trials and reported the effect of TH on the incidence of AKI after perinatal asphyxia [5, 16–23]. Table 1 shows the summary of study characteristics of these studies. Clinical baseline characteristics were similar (p < 0.05) in TH and control groups in 7 studies. In the study by Simbruner et al., the only differences were the temperature at admission and age at randomization, which were both lower in the control group. In the study by Gluckman et al., 5- and 10-minute Apgar scores and background aEEG amplitude were lower in the TH group.

Inclusion and exclusion criteria of the included studies are presented in Table 2. Gestational age of the included neonates varied from $>35–37$ weeks between the studies. Some studies included a minimum birth weight. All studies included several clinical parameters of hypoxic-ischemic injury in their inclusion criteria, including cord gas (range pH $<7.00–7.09$), base deficit (range $>10–16$ mmol/L), 1- ($\leq3$) and 5- (range $\leq5$ - $\leq6$) and 10-minute ($\leq5$) Apgar scores, need for resuscitation, heart rate, oxygen desaturation or arterial oxygen pressure. Exact cut-off values varied slightly between studies. Neurologic findings of neonatal

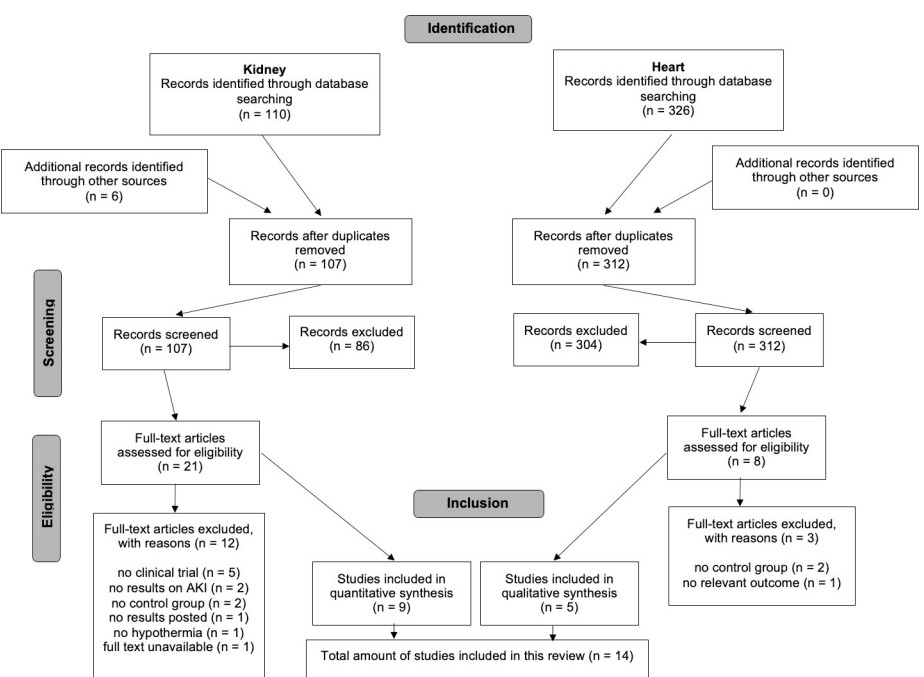

**Fig 1. PRISMA chart of the identification and selection of studies regarding short-term renal and myocardial function.**

encephalopathy, including abnormalities of tone, reflexes, state of consciousness, seizures, posturing and autonomic dysfunction, were also included in most studies. The Sarnat and Sarnat staging was used to assess encephalopathy. An abnormal aEEG was included in two studies.

Four studies included only cases with moderate or severe HIE, of which two studies did not further specify on the ratio between moderate and severe HIE. Four studies also included neonates with mild encephalopathy. One study did not describe the ratio of the different stages of HIE. Table 2 shows the ratios of the severity of HIE.

**Therapeutic hypothermia.** The characteristics of the application of TH for each study are presented in Table 2. All studies initiated the TH within the first 6 hours of life. Four of the included studies applied head cooling with mild or minimal systemic hypothermia and five studies applied whole body cooling.

**Quality assessment.** All studies performed proper randomization and allocation concealment, except for one study in which the mode of allocation concealment was not specified. Because of the impossibility to achieve blinding in these studies, we did not include this criterion in the quality assessment. The assessors of outcome were also not blinded. However, as the outcome is objective, we believe there is still low risk of detection bias. Complete outcome data were reported in seven of the nine trials. Two trials missed data of a few participants. There was no selective reporting. We found possible presence of other bias in six of the nine trials and graded this as 'unclear risk'. In two of these studies this was due to baseline imbalance. In the four other studies this was due to overly wide inclusion criteria for participants as they included all neonates with HIE, including those with mild HIE who may not have benefited from TH. The risk of bias assessment is summarized in Fig 2. Of the nine trials, four were labelled as 'high' quality and five as 'moderate' quality.

**Association between TH and AKI.** All studies examined the incidence of AKI in patients. However, considerable heterogeneity was noted in the definitions used. AKI was defined

**Table 1. Summary of study characteristics of included studies on renal function.**

| Author [ref], study period, country | Study design | Intervention | Number of patients | AKI definition / renal parameters assessed | Time of measurement | Incidence AKI Cooled | Control | Quality assessment |
|---|---|---|---|---|---|---|---|---|
| Akisu et al. [16], 2000–2001, Turkey | Single centre; randomized controlled trial | Head cooling | 11 cooled vs 10 control | ND | ND | 5 of 11 | 5 of 10 | Moderate |
| Eicher et al. [17], ND, Canada; USA | Multicentre; randomized controlled trial | Whole body cooling | 32 cooled vs 33 control | Urine output <1 mL/kg/h, hematuria, creatinine >150 μmol/L | Any time during hospitalization | 2 of 31 | 3 of 31 | Moderate |
| Gluckman et al. [18], 1999–2002, USA; Canada; UK; New Zealand | Cool Cap study; multicentre; randomized controlled trial | Head cooling | 116 cooled vs. 118 control | Urine output <0.5 ml/kg/h for at least 24 h or maximum serum creatinine >90 μmol/L | In first 7 days of life | 73 of 112 | 83 of 118 | Moderate |
| Gunn et al. [19], 1996–1997, New Zealand | Single centre; randomized controlled trial | Head cooling plus either minimal or mild systemic cooling | 12 cooled vs 10 control | Urine output <0.5 ml/kg/h for at least 24 h, proteinuria, hematuria, maximum serum creatinine levels | ND | 12 of 12 | 10 of 10 | High |
| Roka et al. [5], 2005–2006, Hungary | Part of TOBY trial; single centre; randomized controlled trial | Whole body cooling | 12 cooled vs. 9 control | Rate of diuresis and serum creatinine levels | 6h, 24h, 48h, 72h (after birth) | 3 of 12 | 7 of 9 | High |
| Shankaran et al. [20], 2000–2003, USA | NICHD study; multicentre; randomized controlled trial | Whole-body cooling | 102 cooled vs. 106 control | Oliguria or anuria | During hospital course | 16 of 102 | 23 of 106 | High |
| Simbruner et al. [21], 2001–2006, Austria; Germany | Neo.nEURO. network; multicentre; randomized controlled trial | Whole body cooling | 64 cooled vs. 65 control | Urine output <0.5 ml/kg/hour for at least 24h and maximal serum creatinine levels of >90 μmol/L | During intervention period | 16 of 62 | 26 of 63 | High |
| Tanigasalam et al. [22], 2013–2015, India | Single centre; randomized controlled trial | Whole body cooling | 60 cooled vs. 60 control | AKIN criteria: stage 1 (increase in serum creatinine >26.5 μmol/L or serum creatinine >150–200% from baseline), stage 2 (increase in serum creatinine >200–300% from baseline), stage 3 (increase in serum creatinine >300% from baseline or serum creatinine >353.6 μmol/L with an acute rise of >44.2 μmol/L) | 6h, 36h, 72h (after birth) | 19 of 60 | 36 of 60 | High |
| Zhou et al. [23], 2003–2005, China | Multicentre; randomized controlled trial | Head cooling | 100 cooled vs. 94 control | Creatinine >120 μmol/L, blood urea nitrogen >8 mmol/L or urine output <1 mL/kg/h | 12h, 24h, 48h, 72h (after treatment) | 21 of 100 | 19 of 94 | High |

ND = not described.

differently among each of the reported studies. Both serum creatinine and urine output were included in 7 of the studies. One study did not describe the definition used. Table 1 presents the AKI definition used in each study, the time of measurement and the incidence of AKI in the cooled and control groups. Eicher et al., Shankaran et al. and Zhou et al. reported on several renal parameters separately. To calculate the incidence of AKI for this review, serum creatinine was used from the studies by Eicher et al. and Zhou et al. and oliguria from Shankaran et al., as these criteria were most comparable with the other studies.

A total of 504 (range 11–112) neonates were treated with TH in the 9 trials, of whom 167 (33.3%) developed AKI. The control groups consisted of a total of 501 neonates, of whom 212 (42.3%) developed AKI.

**Table 2. Inclusion and exclusion criteria and used intervention in studies on renal function.**

| Study | Inclusion criteria | Exclusion criteria | Severity of HIE in participants (n, %) | | | Intervention | | | | Control |
|---|---|---|---|---|---|---|---|---|---|---|
| | | | Mild | Moderate | Severe | N | Cooling type | Target temperature (˚C) | Period | N |
| **Akisu et al.** [16] | 5-min Apgar score ≤6; severe acidosis; neurologic findings of encephalopathy | Metabolic disorders; congenital malformations; chromosomal abnormalities; congenital infection; transitory drug depression | 3 (14%) | 12 (57%) | 6 (29%) | 11 | Head plus minimal systemic | Ear 33.5–33 / Rectal 36.5–36 | 72 h | 10 |
| **Eicher et al.** [17] | Birthweight >2000g; one clinical indication of hypoxic-ischemic injury; two neurologic findings of neonatal encephalopathy | Maternal chorioamnionitis; sepsis at birth; birth weight or head circumference <10%; presumed chromosomal abnormality | 2 (3%) | 10 (16%) | 50 (81%) | 32 | Whole body | Rectal 33.5 | 48 h | 33 |
| **Gluckman et al.** [18] | 10-minute Apgar score ≤5; continued need for resuscitation; severe acidosis; moderate to severe encephalopathy | Use of anticonvulsants; major congenital abnormalities; head trauma causing major intracranial hemorrhage; severe growth restriction; birthweight <1800g; head circumference <-2 SD; critically ill infants | 0 | ND | ND | 116 | Head plus mild systemic | Rectal 34–35 | 72 h | 118 |
| **Gunn et al.** [19] | Severe acidosis; 5-minute Apgar score ≤6; evidence of encephalopathy | Obvious major congenital abnormalities; metabolic diseases | 0 | ND | ND | 12 | Head plus either minimal systemic (n = 6) or mild systemic (n = 6) | Minimal: rectal 36.3 / Mild: rectal 35.7 | 72 h | 10 |
| **Roka et al.** [5] | 10-minute Apgar score ≤5; continued need for resuscitation at 10min; severe acidosis; moderate to severe encephalopathy; abnormal aEEG | Congenital malformations; suspected metabolic disorders | ND | ND | ND | 12 | Whole body | Rectal 33–34 | 72 h | 9 |
| **Shankaran et al.** [20] | Either severe acidosis or acute perinatal event and 10-minute Apgar score ≤5 or assisted ventilation; moderate or severe encephalopathy | Major congenital abnormality; birth weight of ≤1800 g; moribund infants | 0 | 135 (65%) | 72 (35%) | 102 | Whole body | Esophageal 33.5 | 72 h | 106 |
| **Simbruner et al.** [21] | 10-minute Apgar score <5, continued need for resuscitation, severe acidosis; clinical evidence of encephalopathy; abnormal standard EEG | Use of high-dose anticonvulsant therapy; birth weight <1800 g; head circumference of 3rd percentile; major congenital malformations; imperforate anus; gross hemorrhage; infant "in extremis" | 0 | 41 (33%) | 84 (67%) | 64 | Whole body | Rectal 33.5 | 72 h | 65 |
| **Tanigasalam et al.** [22] | Encephalopathy; severe acidosis; any two of: 10-min Apgar score of ≤5, evidence of fetal distress, assisted ventilation for at least 10 min after birth, evidence of any organ dysfunction | Extramural neonates; major congenital abnormalities; absence of spontaneous respiratory efforts by 20 min after birth; history of maternal renal failure | 5 (4%) | 83 (69%) | 32 (27%) | 60 | Whole body | Rectal 33–34 | 72 h | 60 |

*(Continued)*

**Table 2.** (Continued)

| Study | Inclusion criteria | Exclusion criteria | Severity of HIE in participants (n, %) | | | Intervention | | | | Control |
|-------|-------------------|-------------------|------|----------|--------|-----|--------------|--------------------------|--------|---------|
| | | | Mild | Moderate | Severe | N | Cooling type | Target temperature (˚C) | Period | N |
| Zhou et al. [23] | Birth weight >2500 g; Apgar score ≤3 at 1 minute and ≤5 at 5 minutes; severe acidosis; need for resuscitation or ventilation at 5 minutes of age | Major congenital abnormalities; infection; other encephalopathy; severe anemia (hemoglobin <120g/L) | 39 (21%) | 82 (42%) | 73 (37%) | 100 | Head plus mild systemic | Nasopharyngeal 34 <br> Rectal 34.5–35 | 72 h | 94 |

ND = not described.

The nine trials were included in meta-analysis, presented in Fig 3. There was statistically significant heterogeneity (p = 0.08; $I^2$ = 44%; 95% CI 0–74%). A significant difference between the rate of AKI was observed in cooled infants when compared to the control group (RR = 0.81; 95% CI 0.67–0.98; p = 0.03) with a RD of -0.09 (95% CI -0.16 –-0.01; p = 0.02). Subgroup analysis of the trials that used selective head cooling with mild systemic hypothermia (RR = 0.97; 95% CI 0.86–1.09; p = 0.58) and the trials that used whole-body cooling (RR = 0.58; 95% CI 0.44–0.76; p < 0.0001) demonstrated that the significant effect only applies for the studies using whole-body cooling. The 95% CI of $I^2$ was 0–12% in the subgroup that used head cooling with mild systemic hypothermia and 0–61% in the subgroup that used whole-body cooling. Egger's test was not appropriate to conduct as only nine trials were included. Using the leave-one-out method for the sensitivity analysis, we noticed that leaving out Gluckman et al., Roka et al., Shankaran et al., Simbruner et al. or Tanigasalam et al. separately results in an insignificant result. However, regarding only the subgroup of whole-body cooling, leaving out one of the five trials did not affect significance.

## Myocardial dysfunction

**Systematic search.** Database searching on long-term effects identified 149 citations. Twelve citations were duplicates and 137 studies were included for analysis. However, no relevant studies regarding the long-term effects of TH on the incidence and risk of developing myocardial dysfunction after neonatal asphyxia or other indications for TH were found. Follow-up articles of the CoolCap, NICHD and TOBY trials did not include cardiac parameters [12–14]. Including keywords on other indications for TH in our search identified 109 more studies, but none of these were relevant.

Database searching on short-term effects identified 326 citations. A total of 321 citations were excluded because they were duplicates or the titles and abstracts or full texts were not relevant for this review. Overall, there were 5 studies included in this systematic review [17, 24–27]. A flow diagram is presented in Fig 1.

Studies on the cardioprotective effect of TH in asphyxiated human neonates are limited to the assessment of myocardial function up till 4 days after birth. Five studies were found on these short-term effects. The study of Eicher et al. was included in both the analysis on renal function and myocardial function. An overview of the study characteristics of these studies is presented in Table 3. Two of the studies were an RCT, two were prospective cohort studies and one was a retrospective cohort study. All studies applied TH for 72 hours, except for the study by Eicher et al, which used a period of 48 hours. Inclusion criteria were comparable among the studies.

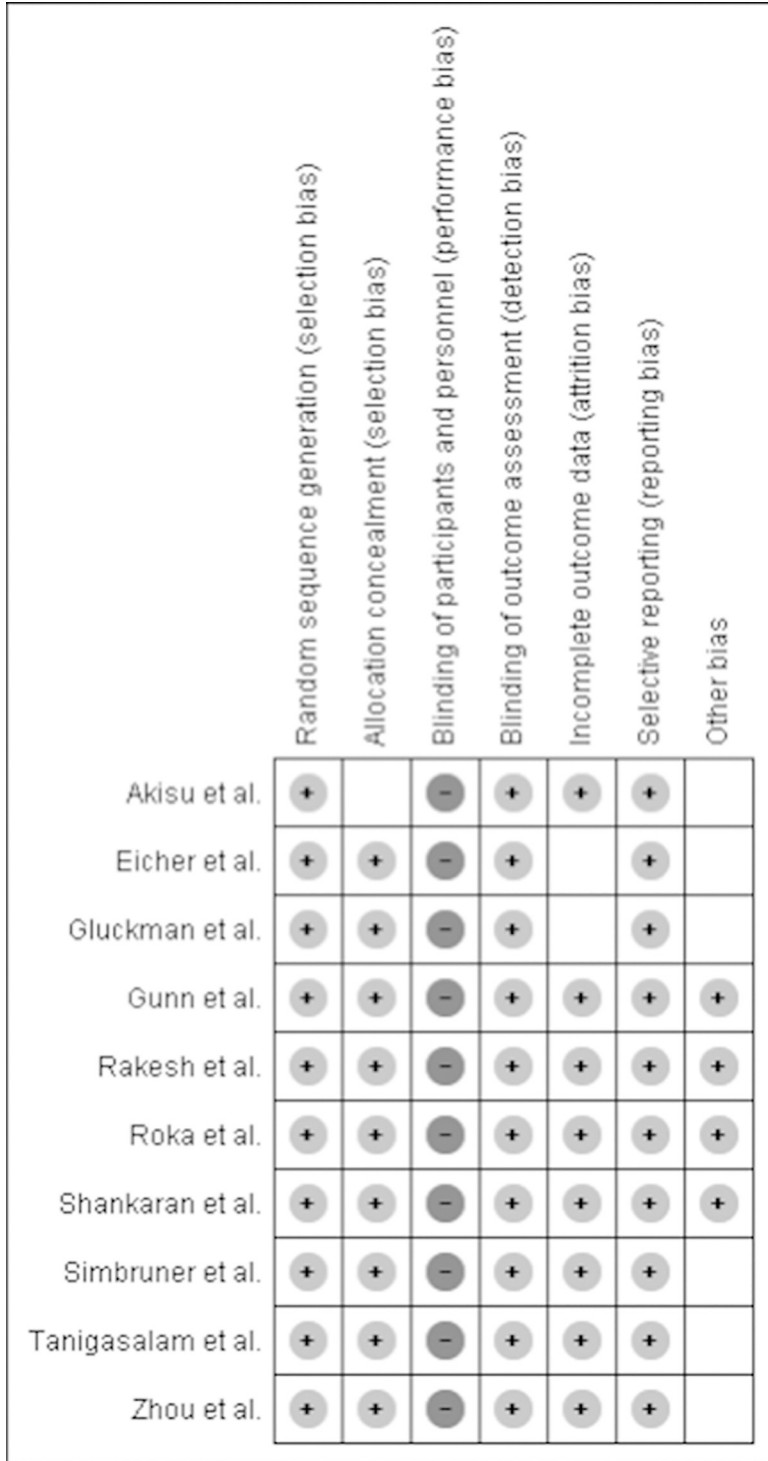

**Fig 2. Risk of bias assessed in the included randomized controlled trials.** Plus-sign represents 'low risk', minus-sign represents 'high risk', empty represents 'unclear risk'.

One of the RCT's was labelled as 'moderate' quality and the other as 'high' quality, summarized in Fig 2. Two of the three observational studies were labelled as 'moderate' quality and one as 'high' quality, presented in Table 4. The third item of the NIHQAT was not applicable

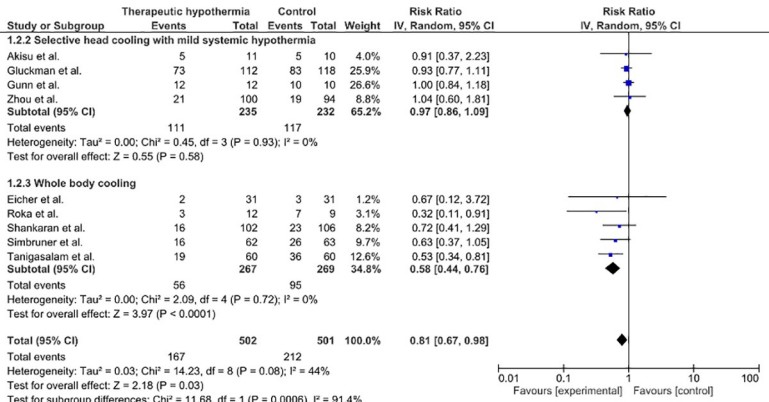

**Fig 3. Forest plot of the effect of hypothermia on the incidence of AKI in asphyxiated neonates, expressed in risk ratios.** Subgroup selective head cooling with mild systemic hypothermia: there is no significant difference between cooled and non-cooled neonates. Subgroup whole body cooling: there is a significant difference between cooled and non-cooled neonates.

for retrospective studies, therefore the needed score for each level of quality was lowered by one. There were significant differences in one or more baseline characteristics between the TH group compared to the control group in the studies by Vijlbrief et al, Liu et al. and Nestaas et al.

**Table 3. Summary of study characteristics and results of included studies on myocardial function.**

| Author [ref], study period, country | Study design | Intervention | Number of patients | Myocardial parameters assessed | Time of measurement | Results | Quality assessment |
|---|---|---|---|---|---|---|---|
| **Eicher et al. [17], ND, Canada; USA** | Multicentre; randomized controlled trial | Whole body cooling | 31 cooled vs. 31 non-cooled | cTnI, CK-MB | 48h | No significant reduction in cTnI; significant increase in CK-MB in hypothermia group | Moderate |
| **Liu et al. [24], ND, UK** | Retrospective cohort study | Whole body cooling (n = 58), head cooling (n = 4) | 61 cooled vs. 14 non-cooled | cTnI | First 3 days | Significant reduction in peak level and AUC of cTnI within 24h in hypothermia group | Moderate |
| **Nestaas et al. [25], 2010–2011, Norway** | Single centre; prospective cohort study | Whole body cooling | 44 cooled vs. 20 non-cooled | Tissue doppler measurements, peak cTnT | Day 1, 3, 4 | Similar impairment during days 1–3, significant improvement in myocardial function in hypothermia group at day 4 (after rewarming) with a better myocardial function than at day 3 in the non-cooled group; higher median peak cTnT in hypothermia group | High |
| **Rakesh et al. [26], 2014–2016, India** | Single centre; randomized controlled trial | Whole body cooling | 60 cooled vs. 60 non-cooled | CK-MB, cTnI, ECG, ECHO | 0, 24 and 72h | Significant increase in median of difference of CK-MB and cTnI in hypothermia group; less findings of myocardial dysfunction on ECG and ECHO at 72h in hypothermia group | High |
| **Vijlbrief et al. [27], 2006–2008, The Netherlands** | Single centre; prospective cohort study | Whole body cooling | 20 cooled vs. 28 non-cooled | cTnI, BNP | 0, 24, 48 and 84h | Significant reduction in BNP at 48 and 84h in hypothermia group; no difference in cTnI between the two groups | Moderate |

ND = not described; cTnI = cardiac troponin I; BNP = brain natriuretic peptide; CK-MB = creatinekinase MB; ECG = electrocardiography; ECHO = echocardiography; AUC = area under the curve; cTnT = cardiac troponin T.

**Table 4. Quality assessment using the National Institute of Health Quality Assessment Tool for observational cohort studies.**

| Study | Q1 | Q2 | Q3 | Q4 | Q5 | Q6 | Q7 | Q8 | Q9 | Q10 | Q11 | Q12 | Q13 | Q14 | Total |
|---|---|---|---|---|---|---|---|---|---|---|---|---|---|---|---|
| Vijlbrief et al. | Y | Y | CD | Y | N | Y | Y | N | Y | NA | Y | N | Y | Y | 9/13Y |
| Liu et al. | Y | N | NA | CD | N | Y | Y | N | Y | NA | Y | N | Y | Y | 7/12Y |
| Nestaas et al. | Y | Y | Y | Y | Y | Y | Y | N | Y | NA | Y | N | Y | Y | 11/13Y |

Q = question; CD = cannot be determined; NA = not applicable; N: = no; Y = yes.

Possible short-term beneficial effects were presented in 4 out of 5 identified studies, assessed by the level of BNP, cTnI and CK-MB and findings of myocardial dysfunction on ECG, ECHO and tissue doppler measurements. An overview of the results of these studies is presented in Table 3. Four of the five studies presented results on the effect of TH on the level of cTnI. Because of the different times of measurements applied in these four studies, no meta-analysis could be performed on the effect of TH on the level of cTnI.

## Discussion

In addition to the brain, perinatal asphyxia exerts a profound harmful effect on the function of major organ systems. The temporary lack of oxygen delivery leading to hypoxic-ischaemic damage can result in multiorgan failure, including cardiovascular and renal dysfunction [5, 28].

In this systematic review we tested the hypothesis that TH, which is used as a neuroprotective strategy, has an additional protective effect on the heart and kidney.

### Renal dysfunction

To evaluate the effect of TH on renal function, we performed two searches on trials that assessed the short-term and long-term effects of TH on the kidneys. However, no long-term follow-up trials on asphyxiated neonates treated with TH that included renal parameters in their outcome were found. Therefore, we performed a review and meta-analysis of studies on the incidence of AKI, assessed during the hospital course in the first period of life, after treatment with TH in comparison to control groups treated without TH. Our meta-analysis showed a significant difference between the rate of AKI in cooled infants and the control group. Furthermore, subgroup analysis showed that the significant effect was only present in the trials that used whole-body cooling. This is likely due to the fact that the renal temperature during head cooling with mild systemic TH did not decrease sufficiently to exert a beneficial effect. At present whole-body cooling is more commonly used, but the superiority of either whole-body cooling or selective head cooling for neuroprotection has not yet been established [29, 30]. Based on the results of this review, whole-body cooling could be more effective than selective head cooling in preventing damage to other organs than the brain, such as the kidneys.

The results of our meta-analysis were slightly different from a Cochrane review on cooling for newborns with HIE including six studies, in which a risk ratio of 0.87 (95% CI 0.74–1.02; P = 0.077) was calculated for renal impairment [31]. This Cochrane review showed a superior effect of whole-body cooling over selective head cooling as well.

With the redistribution of cardiac output as a reaction to asphyxia, blood is directed preferentially to the brain and the heart, thereby limiting oxygen delivery to the kidneys. Renal cells only have a limited capacity for anaerobic respiration, as the tubular cells already live in an environment of low oxygen tension, and are highly susceptible to reperfusion injury [2, 6]. As

a result, AKI is frequently seen in asphyxiated neonates. With an incidence of 40–50%, renal injury is a common organ dysfunction after perinatal asphyxia [22]. AKI is characterized by a period of impairment in the kidney's excretory function. The AWAKEN study found an overall incidence of 36.7% of AKI in neonates, including those with HIE, born at a gestational age of ≥36 weeks. This study also determined that AKI is an independent risk factor for mortality and longer hospital stay [32]. Previous studies have shown an incidence of 36.1%-72% of AKI after perinatal asphyxia, which is comparable to the incidence we found in the pooled control groups (42.3%) [33–40]. Furthermore, studies have found that AKI correlates with the severity of asphyxia, mortality and neurological outcome [33, 41]. A possible long-term consequence is nephron loss, causing hypertension and proteinuria, which in turn leads to progressive renal damage. Hyperfiltration in the remaining nephrons, together with an impairment in renal oxygenation due to capillary rarefaction, proteinuria as a result of glomerular damage, chronic overactivity of the renin-angiotensin-aldosteron system (RAAS) and interstitial inflammation all contribute to the progression of CKD [42, 43]. Multiple animal studies have shown that ischemia can cause permanent kidney damage as a result of fibrosis, inflammation and loss of peritubular capillaries [44–46].

A systematic review and meta-analysis by Greenberg et al. evaluated the long-term risk of CKD and mortality after an episode of AKI in children [47]. Ten cohort studies evaluating long-term renal outcomes were selected. Included in these studies were a total of 346 patients with a mean follow-up of 6.5 years (range 2–16). They determined the cumulative incidence rate of hypertension, proteinuria, GFR <90 ml/min/1.73m$^2$, GFR <60 ml/min/1.73m$^2$, end stage renal disease and mortality per 100 patient-years, which respectively were 1.4, 3.1, 6.3, 0.8, 0.9 and 3.7. However, these studies used a total of six different definitions of AKI and were of variable quality. There was a considerable difference in outcome between the studies, presumably as a result of discordant outcome measures, study size and differences in methodology. There was no control group without AKI included in any of these studies. However, studies in adults did include control groups and hereby demonstrated that AKI acts as an independent risk factor for CKD [48].

An important limitation of our review is the difference in definitions of AKI used in the included studies, therefore caution before interpreting these findings is advised. The KDIGO guideline uses a definition of an increase in serum creatinine of ≥26.5 μmol/L or ≥50% within 48 hours or urine output of <0.5 mL/kg/hour for >6 hours [49]. However, in newborns the serum creatinine concentration normally decreases over the first days of life, urine production is notoriously difficult to determine accurately and because of limited renal concentration capacity, renal osmolar excretion can already be insufficient at urine production rates below 1.5mL/kg/hour [43]. For this review, presence of oliguria was used in the definitions of five of the included studies and increased creatinine levels in seven studies. In the study by Gunn et al., neonates with proteinuria or hematuria were also considered to suffer from AKI. This study reported signs of AKI in all participating neonates, which is likely a result from the use of a too broad definition. We believe the difference in results between the studies can presumably be partially explained by these discrepant definitions. A universal definition of neonatal AKI is important to enable reliable comparison of different trials. Several studies have proposed neonatal AKI definitions using modifications to the KDIGO definition and RIFLE criteria [43, 50].

Other limitations are small study sizes in some studies (<100 participants in 4 studies) and different proportions of the severity of HIE between the studies. Four studies did also include neonates with mild HIE, who may not have benefited from TH, while other studies only included moderate and severe HIE. Thus far, neither the safety nor efficacy of TH for mild HIE has been demonstrated [51]. However, for the development of AKI, no subgroup analysis

on the severity of HIE was performed in these studies. More severe HIE might also imply more severe damage in other organs. Included studies in this review found that a more severe stage of HIE results in a higher rate of mortality and severe disability. Gluckman et al. found no apparent improvement on mortality and severe disability with TH in neonates with the most severe or most advanced aEEG changes after birth. Regarding the effect of TH on the incidence of AKI in infants with mild HIE, the two studies with respectively 14% and 21% neonates with mild HIE used head cooling, so the fact that these studies did not show a superior effect is likely due to the method of cooling. The two other studies, that used whole body cooling, only included respectively 3% and 4% neonates with mild HIE, which is too little to draw any conclusions on the effect of TH on AKI in case of mild HIE.

Another limitation is the fact that the presence of publication bias can only be reliably assessed when there are at least 10 studies included in the meta-analysis [7, 52]. With only nine trials we can therefore not rule out the presence of publication bias in our meta-analysis. Furthermore, included studies were performed between 1996 and 2015. More recent studies are potentially more relevant, since there is lower risk of publication bias and adjustments in clinical practice over time could have affected the outcomes [53]. However, as TH has been the standard of care in developed countries for the treatment of neonates with moderate to severe HIE, relatively recent studies comparing this intervention to a control group without TH are rare [4].

In our review, we found an $I^2$ value of 44%. According to the Cochrane Handbook, a $I^2$ value of 0–40% might not be important, a value of 30–60% may represent moderate heterogeneity, a value of 50–90% may represent substantial heterogeneity and a value of 75–100% may represent considerable heterogeneity [7]. The $I^2$ value we found in our meta-analysis may represent moderate heterogeneity. This is most likely due to differences in method used, as the heterogeneity in the two subgroups were both 0%. However, the $I^2$ statistic is underpowered to detect heterogeneity with a small number of studies, so the presence of heterogeneity in these subgroups cannot be ruled out [54]. The calculated confidence intervals also show that some heterogeneity might be present in the subgroups.

It is important to also keep in mind other risk factors that could influence renal function and increase the risk for developing AKI other than HIE. Six of the nine studies provided data on the incidence of hypotension and in all these studies hypotension was equally present in the hypothermia groups and control groups. Other risk factors for AKI were not part of the exclusion criteria and were not adjusted for, which is another possible limitation of this review.

Overall, it is likely that neonates who suffered from AKI are at higher risk of developing CKD later in life. Consequently, reducing the incidence of AKI in asphyxiated neonates is of great importance in preventing the progression to CKD. Therefore, as TH resulted in a significantly lower risk of developing AKI in our meta-analysis, we argue that this treatment yields a protective effect on the long-term function of the kidneys. Furthermore, as AKI is a risk factor for developing CKD and early treatment of risk factors for rapid progression like hypertension and proteinuria protect renal function, we recommend screening on the development of CKD throughout life in children with a history of AKI in the neonatal period, which is currently not performed [55].

## Myocardial dysfunction

To evaluate the effect of TH on myocardial function, we performed a search on trials that assessed the long-term effect of TH on the heart. However, no long-term follow-up trials were found on asphyxiated neonates treated with TH that included cardiac parameters in their outcome. Results from short-term studies suggest a trend towards a decrease in cardiac

biomarkers and myocardial dysfunction assessed by ECG, ultrasound and tissue-Doppler in asphyxiated neonates treated with TH compared to normothermia. Whether this short-term cardioprotective effect also implies a positive effect on the long-term must be explored in further research. Furthermore, animal studies also suggest a cardioprotective effect of TH. Treatment with TH in hypoxic-ischaemic newborn pigs significantly reduced pathological cardiac lesions and the cardiac biomarker troponin I [56]. Another animal study on embryonic rat hearts after oxidative stress has shown a cardioprotective role of TH by reducing cardiomyocyte injury [57]. The possible positive effect of TH on myocardial dysfunction is likely a result of a reduction in cardiac metabolism, cardiac output and oxygen demand during TH [26].

The five studies included different cardiac parameters and the time of measurement of these parameters also differed between the studies. BNP levels correlate well with echocardiographic measurements of myocardial dysfunction [27]. The sensitivity and specificity of CK-MB is lower than that of cTnI and cTnT as it is influenced by other factors, like kidney injury, gestational age, mode of delivery and birth weight [58]. cTnI levels correlate well with the degree of myocardial damage in asphyxiated neonates and might be an appropriate marker of the anticipated severity of myocardial dysfunction [59]. A systematic review by Teixeira et al. investigating the use of cardiac biomarkers in neonates found that cTnI and cTnT levels are useful tools for assessing myocardial dysfunction in asphyxiated neonates, with a higher sensitivity and specificity than echocardiography and other biomarkers. However, cTnI and cTnT can also be elevated in congenital heart defects (only cTnT), patent ductus arteriosus (only cTnT) and respiratory distress [58].

Changes in ECG, performed after the first 24 hours of life, and echocardiography including Doppler tissue imaging have also been proven to be reliable markers [25, 60]. Approximately 30% to 50% of asphyxiated neonates show indications of ventricular dysfunction on echocardiography [61].

cTnI and cTnT appear to be the most reliable biochemical markers for myocardial dysfunction.

Studies on the effect of TH after myocardial infarction in adults showed a trend towards a reduction in infarct size (IS) [62–70]. However, this reduction was only statistically significant in the IS normalized to myocardium at risk in two of the nine trials, which were the only two trials in which all patients reached a temperature of <35˚C before reperfusion. An absolute reduction in IS of 5% is currently accepted as a clinically meaningful result for cardioprotective trials [67]. This reduction was achieved in four of the nine trials. The cardioprotective effect of TH appeared to be very dependent on reaching a temperature of <35˚C before reperfusion took place. However, it is not clear how well these results from studies on myocardial infarction may be extrapolated to asphyxiated neonates because of the marked differences in clinical picture. In myocardial infarction, the cardiac damage is focal, due to ischemia and often a result from pre-existing coronary artery disease. Furthermore, adult hearts and neonatal hearts might react differently to damage and it is possible that neonatal hearts are more resilient and salvageable.

Myocardial function and its markers can be influenced by several factors. Nestaas et al. could not assess the impact from the use of inotropic medication on myocardial function as this is probably more often used in infants in an impaired hemodynamic state. Liu et al. investigated the effect of cardiovascular confounders on the peak level of cTnI before initiation of cooling. Receiving cardiac compressions was the only near significant variable (p = 0.06). Receiving adrenaline had no effect on the level of cTnI shortly after resuscitation. Vijlbrief et al. found no significant correlations between cTnI and BNP levels and possible confounders, such as epinephrine use at resuscitation, changes in heart rate, pulmonary hypertension (PPHN) or hypotension treatment. Several included studies in this review excluded neonates

with sepsis at birth or with major congenital abnormalities. Other risk factors for myocardial ischemia were not part of the exclusion criteria and were not adjusted for, which is another possible limitation of this review.

### Future research

There appears to be an important gap in literature concerning the long-term effect of TH on renal and myocardial function in asphyxiated neonates. This review showed a renoprotective effect to be very likely. However, additional research will be needed to confirm the positive effect of TH on long-term renal function. Research should evaluate the risk of developing CKD and heart failure later in life in in children with a history of perinatal asphyxia treated with or without TH. Determining whether neonatal hypothermia reduces chronic organ dysfunction will require long-term follow-up of the children in RCT's. However, as TH is now the standard of care in perinatal asphyxia, it will be difficult to compare the effect of TH to a control group in new studies outside of historic controls. Therefore, follow-up studies of large trials on the effect of TH in perinatal asphyxia, e.g. NICHD and TOBY, should also assess renal and cardiac parameters, like creatinine, proteinuria, blood pressure and echocardiography. Furthermore, a sizeable proportion of neonates with asphyxia still develops AKI even after cooling. This supports research into the development of other therapeutic options to further decrease renal injury. For example, treatment with theophylline was found to reduce the incidence of AKI in term neonates with severe asphyxia at birth by 60%. However, the effect of this treatment has not been evaluated in neonates receiving TH [71]. For clinical reasons, follow-up of asphyxiated neonates with renal or cardiac involvement should include assessment of renal and cardiac parameters. In general, the focus of follow-up visits of these children is on neurodevelopment. Future studies should also assess the best methods to assess renal and cardiac function in these children. During follow-up, renal function can be assessed with parameters like creatinine, proteinuria and blood pressure and cardiac function with echocardiography including Doppler tissue imaging. This is both relatively inexpensive and easy and the development of CKD and heart failure can be prevented or timely treated to avoid further costs and minimize the burden of disease for these patients.

## Conclusion

In conclusion, systemic TH had definite protective effects on renal and probably also myocardial function in the days following perinatal asphyxia in (near) term infants. Further studies are needed to describe the long-term effects of systemic TH on renal and myocardial function.

## Supporting information

**S1 Table. PRISMA checklist.**
(DOC)

**S1 Text. Full search strategy.**
(DOCX)

**S2 Text. Items included in the National Institute of Health Quality Assessment Tool (NIH-QAT).**
(DOCX)

## Author Contributions

**Conceptualization:** Maureen van Wincoop, Karen de Bijl-Marcus, Floris Groenendaal.

**Formal analysis:** Maureen van Wincoop, Agnes van den Hoogen.

**Investigation:** Maureen van Wincoop.

**Methodology:** Maureen van Wincoop, Karen de Bijl-Marcus, Floris Groenendaal.

**Supervision:** Karen de Bijl-Marcus, Floris Groenendaal.

**Writing – original draft:** Maureen van Wincoop.

**Writing – review & editing:** Karen de Bijl-Marcus, Marc Lilien, Agnes van den Hoogen, Floris Groenendaal.

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
