## [Decision Letter · Decision Letter 0]

10 Dec 2020

PONE-D-20-36313

The long-term effects of therapeutic hypothermia on renal and myocardial function in asphyxiated full-term neonates: a systematic review and meta-analysis

PLOS ONE

Dear Dr. de Bijl-Marcus,

Thank you for submitting your manuscript to PLOS ONE. After careful consideration, we feel that it has merit but does not fully meet PLOS ONE’s publication criteria as it currently stands. Therefore, we invite you to submit a revised version of the manuscript that addresses the points raised during the review process.

We look forward to receiving your revised manuscript.

Kind regards,

Karel Allegaert

Academic Editor

PLOS ONE

Journal Requirements:

2. Please provide an analysis of publication bias also known as sensitivity analysis on the studies included in the systematic review.

Additional Editor Comments (if provided):

the paper has been assessed by 3 clinical reviewers and one 'statistical/methods' reviewer

all value the effort, but have raised issues that should be further considered.

Reviewers' comments:

Reviewer's Responses to Questions

**Comments to the Author**

1. Is the manuscript technically sound, and do the data support the conclusions?

Reviewer #1: Partly

Reviewer #2: Yes

Reviewer #3: Yes

Reviewer #4: Yes

2. Has the statistical analysis been performed appropriately and rigorously? 

Reviewer #1: Yes

Reviewer #2: Yes

Reviewer #3: Yes

Reviewer #4: Yes

3. Have the authors made all data underlying the findings in their manuscript fully available?

Reviewer #1: Yes

Reviewer #2: Yes

Reviewer #3: Yes

Reviewer #4: Yes

4. Is the manuscript presented in an intelligible fashion and written in standard English?

Reviewer #1: Yes

Reviewer #2: Yes

Reviewer #3: Yes

Reviewer #4: Yes

5. Review Comments to the Author

Reviewer #1: I will focus on methods and reporting.

Quallity assessment is appropriate.

Major

1) Avoid fixed effect models since they under-perform in the presence of ANY heterogeneity. Random-effects (RE) models are more conservative and provide better estimates with wider confidence intervals: http://www.ncbi.nlm.nih.gov/pubmed/11252006 and http://www.ncbi.nlm.nih.gov/pubmed/21148194 . What does the arbitrary 50% cut-off point add? Why is 49% fine and 51% an issue? My point is that a RE model will work better, in the presence of 5% heterogeneity, compared to a FE model!

2) Report the confidence intervals for I^2 (calculated using heterogi or metaan in Stata) as argued in http://www.ncbi.nlm.nih.gov/pubmed/17974687. A simple formula exists in the seminal 2002 Higgins paper that proposed I^2.

Minor

1) Abstract: meta analysis methods should be included in the abstract i..e random or fixed effects modelling, how was heterogeneity quantified, publication bias assessed etc.

2) Do not use the term WMD, it is confusing. Say MD instead. Please see http://handbook.cochrane.org/chapter_9/9_2_3_1_the_mean_difference_or_difference_in_means.htm

3) I^ is not a test but a statistic

4) Publication bias tests and plots only relevant if you have >10 studies otherwise underpowered to detect much and tend to lead to conclusions that are not justified http://www.ncbi.nlm.nih.gov/pubmed/11106885. If you don’t have enough studies to assess you should discuss this as a major limitation. Even with 10 or 20 studies it is very difficult to visually assess. If you have 20 or more studies it is a considerable strength.

5) Year may be worth considering in bias assessment, especially if you don't have enough studies for a formal test: http://www.ncbi.nlm.nih.gov/pubmed/25988604. With newer studies we would be more confident.

6) Note that MH is traditionally a fixed effect approach and the random effects version in RevMan is an IV-MH hybrid method. Some clarification on the MH weighting is needed (it is inverse variance as well, like your analysis of the continiuous outcomes).

7) How was the random-effect model implemented, i.e. how was heterogeneity estimated? There are numerous ways to do so. Did they use the standard DerSimonian-Laird method? If so, please state so. Also there are better performing methods, for example please see https://www.ncbi.nlm.nih.gov/pubmed/28815652 (or http://www.ncbi.nlm.nih.gov/pubmed/23922860) and the metaan command in Stata where these are implemented (https://www.stata-journal.com/article.html?article=st0201).

8) Cochran Q (i.e. chi-square) is notoriously underpowered to detect heterogeneity, especially for small meta-analyses http://www.ncbi.nlm.nih.gov/pubmed/9595615. I would not use

9) Regarding heterogeneity estimates, all these estimates are very likely off, especially for small meta-analyses, and you should be wary about homogeneity assumptions http://www.ncbi.nlm.nih.gov/pubmed/23922860.

Reviewer #2: Comments on the manuscript # PONE-D-20-36313 entitled «The long-term effects of therapeutic hypothermia on renal and myocardial function in asphyxiated full-term neonates: a systematic review and meta-analysis»

This is a really interesting systematic review and meta-analysis on the possible protective effect on renal and myocardial function of infants undergoing TH. According to the results, TH in asphyxiated neonates reduces the incidence of AKI, thus potentially being renoprotective and, also, might have a cardioprotective effect on the short-term.

In general, this is a well written paper and, I only have minor comments. The language is satisfactory.

Title

I think that the title should change to «Effect of therapeutic hypothermia on renal and myocardial function in asphyxiated neonates: a systematic review and meta-analysis».

The reasons why I am suggesting this change is due to the fact that a) the impact of TH on myocardial dysfunction was not made possible to be evaluated in the long-term, b) mostly short-term outcomes were studied, and c) TH includes both near-term and term infants (this should be taken into consideration in the whole manuscript).

Abstract

1. Objective: it should be: to evaluate the short- and long-term effects of TH on renal and myocardial function in asphyxiated neonates.

Introduction

1. L76. Please add the “possible” short- and long-term effects.

Methods – Search strategy

1. L94: ….to possibly identify a valid predictor…! I am not sure what the meaning of this statement is! Do authors try to identify any predictor of renal or cardiac dysfunction? Please clarify!

2. L105: A relevant comment regarding criterion # 5! Please, rephase or clarify!

3. L113: Please change reviewer to investigator of researcher.

Results

1. L146-147: Please, delete the phrase “Two different ….” As this is already mentioned in the method section.

2. L172: it is written that “There were no significant differences in baseline characteristics between the TH group compared to the control group in 7 studies”. How is it concluded? Please, delete or rephrase the statement.

3. L267: As before, delete the sentence regarding the 2 searchers…

4. L320: Apparently the aim of the study was to assess both short- and long-term effects of TH on the kidneys.

Discussion

It does provide a satisfactory explanation of the results, although it could be slightly shortened.

1. L 322: it is said that “…a review and meta-analysis of studies on the incidence of AKI after treatment with TH was performed”. Generally speaking, I believe that the terms “short- and long-term” should be more clearly specified in the present document (in possibly in the method?). Therefore, at this point it would be better to state “the incidence of AKI during the first week of life” instead to after treatment which is rather indefinite and general.

2. L352: what does RAAS stand for?

Reviewer #3: Thank you for the invitation to review the Manuscript PONE-D-20-36313

"The long-term effects of therapeutic hypothermia on renal and myocardial function in asphyxiated full-term neonates: a systematic review and meta-analysis"for PLOS ONE.

In this systematic review, the authors reported the short- and long-term effects of therapeutic hypothermia (TH) on renal and myocardial function in asphyxiated full-term neonates. Therefore, the data was obtained from an electronic search strategy incorporating MeSH terms and keywords in October 2019 and updated in June 2020 using PubMed and Cochrane databases. Inclusion criteria consisted of an RCT or observational cohort design.

They mainly identified nine studies which described the effect of TH on the incidence of acute kidney injury (AKI) after perinatal asphyxia. Meta-analysis showed a significant difference between the incidence of AKI in neonates treated with TH compared to the control group.

The authors concluded that TH in asphyxiated neonates reduces the incidence of AKI. No studies were found which investigated the effects of TH on long-term renal function or myocardial function. Only few studies suggest short-term beneficial effects on myocardial function.

General comments

The study is generally well written and methodologically sound.

The abstract has a clear objective. The introduction is clear and the aim is well defined. The main strength of the study is that the authors report extensively on the search strategy, study selection and quality assessment with the main result of protective effects on renal function in 9 studies in TH treated asphyxiated neonates which were included in the meta-analysis.

The weakness of the systemic review is that the authors did not find any studies which describe the long-term effects on renal or myocardial function. I'm not convinced that the description of the values in the text adds to what's already presented in the Figures, in fact there's unnecessary repetition between text and figures.

Furthermore, I am wondering that the authors did not consider blood pressure and the use of catecholamines or volume therapy when assessing kidney function

I have several specific concerns

To title:

The title should be changed as no studies were found investigating the long-term effects.

To results section:

The text in the results section includes many redundant data which are already provided in Figures 1 and 3. The authors should avoid repeating the same information.

Page 9, line 179: the title of table 1 is not complete.

Page 12, line 186: the value of the range pH <7.0-7.9 should be checked.

To discussion section:

As mentioned in general comments, the authors should expand the discussion section about the influence of blood pressure and volume therapy on renal function in asphyxiated neonates treated with TH.

Furthermore, they should add some more limitations of the systemic review.

For example, which other factors can influence the renal and myocardial function.

Moreover, it may be very helpful for long-term outcome studies, e.g. assessment of myocardial function by echocardiography should be specified.

In conclusion, the paper presents an interesting review about short-term protective effects of therapeutic hypothermia (TH) on renal and probably myocardial function in asphyxiated full-term neonates. The data can be useful for physicians caring for asphyxiated neonates.

Reviewer #4: Van Wincoop, et al performed a systematic review and meta-analysis of RCT and observational cohort studies of infants with perinatal asphyxia to identify the short and long-term effects of therapeutic hypothermia on renal and myocardial function. They identified 9 studies meeting criteria for short term renal dysfunction (acute kidney injury) and 5 studies meeting criteria for short term myocardial dysfunction. No long-term studies were identified. Meta-analysis found that TH reduced incidence of AKI, with suggestion that TH also was cardioprotective for short term outcomes.

This was a well-written systematic review and meta-analysis of the ongoing problem of multiorgan dysfunction for hypoxic ischemic encephalopathy. It was great that the authors recognize the lack of long-term follow-up studies in this population for systems other than neurodevelopment. However, they should further emphasize that therapeutic hypothermia is now considered standard of care in developed countries for moderate to severe HIE, and it will be difficult to compare outcomes to a non-TH group outside of historic controls. We may have a limited window to concurrently investigate longer-term outcomes in children.

One concern with this study is that the rationale for selecting renal and cardiac dysfunction was not well delineated. It seems like the myocardial dysfunction aspect was tacked on and was not as well explored. The manuscript therefore was not as cohesive. Renal and cardiac manifestations after birth asphyxia may be very disparate. In fact, one could argue that cardiac dysfunction could incorporate pulmonary hypertension, poor cardiac output, or evidence of ischemia. Wonder if it would be better to tackle only renal dysfunction in this analysis and focus a separate manuscript on a more detailed cardiac analysis? Or at least justify why these two end-organs were explored and provide more robust assessment of cardiac portion.

A few other specific comments:

• Unclear why search was expanded to include other indications for TH such as near-drowning and aortic arch surgery when inclusion criteria were only for patients in the setting of perinatal asphyxia.

• Long time span of included studies (between 1996-2015). Limitation may be changes in clinical practice over time which could have affected outcomes.

• Good discussion of other limitations- especially heterogeneity of definition of AKI and differences that might be anticipated by including babies with mild HIE into analyses (sensitivity analysis done)

• As previously mentioned, the myocardial dysfunction discussion needs more detail. Not all studies included all markers for myocardial dysfunction and there seems to be very heterogeneous definition of dysfunction as measured by CK-MB, troponin, BNP, ECG and non-specific ECHO measures in one study, and tissue-Doppler in another study. What is the best marker/definition? Heart failure is different than biochemical or electrocardiographic evidence of ischemia.

• Results from myocardial dysfunction analysis states “Possible short-term beneficial effects were presented in 4 out of 5 identified studies”. While I understand the descriptive nature given no unified diagnosis of myocardial dysfunction, it is difficult to compare studies. I notice that 4/5 studies did investigate cardiac troponin I levels. Perhaps, at least this measure could have been quantitatively compared between studies?

• Similarly in the abstract, it states “Possible short-term beneficial effects were presented in 4 out of 5 identified studies” for myocardial function. This is vague and should either be expanded on or left out entirely.

• Introduction, line 55. Should read “1 to 2 per 1000” instead of “1 to 2 per 1.000”

6. PLOS authors have the option to publish the peer review history of their article (what does this mean?). If published, this will include your full peer review and any attached files.

Reviewer #1: No

Reviewer #2: No

Reviewer #3: No

Reviewer #4: No

---

## [Author Response · Author response to Decision Letter 0]

23 Jan 2021

Please see the rebuttal letter we uploaded for our responses to the comments of the reviewers and editor.

---

## [Decision Letter · Decision Letter 1]

8 Feb 2021

Effect of therapeutic hypothermia on renal and myocardial function in asphyxiated (near) term neonates: A systematic review and meta-analysis

PONE-D-20-36313R1

Dear Dr. de Bijl-Marcus,

We’re pleased to inform you that your manuscript has been judged scientifically suitable for publication and will be formally accepted for publication once it meets all outstanding technical requirements.

Kind regards,

Karel Allegaert

Academic Editor

PLOS ONE

Additional Editor Comments (optional):

no additional comments

Reviewers' comments:

Reviewer's Responses to Questions

**Comments to the Author**

1. If the authors have adequately addressed your comments raised in a previous round of review and you feel that this manuscript is now acceptable for publication, you may indicate that here to bypass the “Comments to the Author” section, enter your conflict of interest statement in the “Confidential to Editor” section, and submit your "Accept" recommendation.

Reviewer #1: All comments have been addressed

Reviewer #2: All comments have been addressed

Reviewer #3: All comments have been addressed

Reviewer #4: All comments have been addressed

2. Is the manuscript technically sound, and do the data support the conclusions?

Reviewer #1: Yes

Reviewer #2: Yes

Reviewer #3: Yes

Reviewer #4: Yes

3. Has the statistical analysis been performed appropriately and rigorously? 

Reviewer #1: Yes

Reviewer #2: Yes

Reviewer #3: Yes

Reviewer #4: Yes

4. Have the authors made all data underlying the findings in their manuscript fully available?

Reviewer #1: Yes

Reviewer #2: Yes

Reviewer #3: Yes

Reviewer #4: Yes

5. Is the manuscript presented in an intelligible fashion and written in standard English?

Reviewer #1: Yes

Reviewer #2: Yes

Reviewer #3: Yes

Reviewer #4: Yes

6. Review Comments to the Author

Reviewer #1: Thank you, I think the authors have addressed all comments. Previously, I was not suggested that the MH hybrid was worse performing that DL IV - as far as I know there is no direct comparison. I was suggesting a clarification on weighting. Anyway, I'm happy with using DL IV instead.

Reviewer #2: None. I think that the authors have made all suggested corrections, and that the article is ready for publication.

Reviewer #3: The manuscript is greatly improved. The changes in the revised manuscript are in accordance with the recommendations and the paper now could be suitable for publication.

Reviewer #4: Van Wincoop, et al have submitted a revision to their systematic review and meta-analysis to identify the effects of therapeutic hypothermia on renal and myocardial function. The authors have made appropriate revisions and have addressed all my concerns.

7. PLOS authors have the option to publish the peer review history of their article (what does this mean?). If published, this will include your full peer review and any attached files.

Reviewer #1: No

Reviewer #2: No

Reviewer #3: No

Reviewer #4: **Yes: **Valerie Chock

---

## [Editor Report · Acceptance letter]

12 Feb 2021

PONE-D-20-36313R1 

Effect of therapeutic hypothermia on renal and myocardial function in asphyxiated (near) term neonates: A systematic review and meta-analysis 

Dear Dr. de Bijl-Marcus:

I'm pleased to inform you that your manuscript has been deemed suitable for publication in PLOS ONE. Congratulations! Your manuscript is now with our production department. 

Kind regards, 

on behalf of

Dr. Karel Allegaert 

Academic Editor

PLOS ONE